# Insights into ground strike point properties in Europe through the EUCLID Lightning Location System

Dieter Roel Poelman[1], Hannes Kohlmann[2], Wolfgang Schulz[2]

[1]Royal Meteorological Institute of Belgium, Brussels, Belgium
5   [2]Austrian Lightning Detection and Information System (ALDIS), Vienna, Austria

*Correspondence to*: Dieter R. Poelman (dieter.poelman@meteo.be)

**Abstract.** Evaluating the risk of lightning strikes to a particular structure typically involves adhering to the guidance outlined in IEC 62305-2. Among the multitude of factors influencing the overall risk, flash density emerges as a crucial parameter. According to its definition, each flash is assigned only one contact point to ground. Nevertheless, it is well known that, on 10   average, flashes exhibit multiple ground termination points as shown by high-speed camera observations. In this research, lightning data collected by the European Cooperation for Lightning Detection (EUCLID) network is utilized in combination with a ground strike point (GSP) algorithm that aggregates individual strokes within a flash into ground strike points. This approach enables the examination of spatio-temporal patterns of GSPs across Europe throughout a decade, spanning from 2013 to 2022. Average GSP densities exhibit variations across the European continent, mirroring the observed patterns in flash 15   densities. The highest densities are concentrated along the Adriatic Sea and the western Balkan region, reaching peak values of up to 8.5 GSPs km$^{-2}$ yr$^{-1}$. The spatial distribution of the mean number of ground strike points per flash reveals a noticeable increase in the Mediterranean, Adriatic, and Baltic Sea regions compared to inland areas. Moreover, it has been determined that the average number of GSPs per flash reaches its peak between September and November. Additionally, a daily pattern is discernible, with the lowest number of GSPs per flash occurring between 12 and 18 UTC (Universal Time Coordinated). It is 20   found that 95% of the separation distances between distinct GSPs are less than 6.7 km. Lastly, it is worth noting that the presence of the Alps has an impact on GSP behaviour, resulting in lower GSP counts in comparison to the surrounding areas, along with the shortest average distances between different GSPs.

## 1. Introduction

In order to make informed decisions about lightning protection measures to a particular structure, it is common practice to 25   follow the guidelines set out in IEC 62305-2 (IEC 62305-2, 2010), which serves as the authoritative standard for lightning risk assessment for structures. Among the various components that influence the risk estimation, the standard puts forward the flash density, $N_G$, representing the number of lightning flashes per square kilometer per year, as one of the key parameters. However, by definition, the location of a flash is determined by the position of the first cloud-to-ground (CG) stroke within the flash. On the other hand, numerous studies, supported by high-speed camera observations (Rakov et al., 1994; Valine et

al., 2002; Saraiva et al., 2010; Poelman et al., 2021a), have provided evidence that, on average, multiple ground strike points (GSPs) exist within multiple-stroke flashes. Hence, GSP densities, $N_{SG}$, should be given the pivotal role in lightning studies, particularly in the context of assessing lightning-related risks. The use of GSP densities is facilitated by the existence of modern lightning location systems (LLSs) that can accurately pinpoint most, if not all, individual strokes within a lightning flash, with precision typically within a few hundred meters. Furthermore, studying variations in GSP densities over time and across

geographical regions aids in the development of comprehensive lightning risk maps, contributing to improved lightning protection standards and guidelines and consequently ensuring the safety of people and property.

A recent study Poelman et al. (2021b) delved into the performance of GSP algorithms in accurately identifying ground strike points through high-speed camera observations. The latter study found that these algorithms performed well, achieving success rates of up to approximately 90% in correctly identifying stroke types within a flash, whether they create new termination

points or follow pre-existing channels. However, for certain applications, the accurate determination of the total number of GSPs is of greater importance. In such cases, misclassifications in stroke types may offset each other and still yield accurate results in terms of the total number of GSPs. A subsequent investigation conducted by Poelman et al. (2023) explored the relationship between the count of GSPs obtained through a specific GSP algorithm and the number of GSPs found in various ground truth datasets. This analysis encompassed four distinct GSP algorithms and provided insights into whether a specific

GSP algorithm tends to overestimate or underestimate the count of ground strike points. It was revealed that each of the four GSP algorithms exhibited some degree of overestimation or underestimation regarding the GSP count per flash. However, the calculated numbers were reasonably close, i.e., within 10 %, to the values observed in ground-truth observations.

In this paper, lightning data of negative CG strokes are gathered from the European Cooperation for Lightning Detection (EUCLID) network and ingested in a ground strike point algorithm in order to analyse spatial and temporal patterns of GSPs

across Europe from 2013 to 2022. Section II provides a description of the LLS data and GSP algorithm employed. Section III then examines the spatio-temporal GSP characteristics within the EUCLID domain. Our findings and a summary are offered in Section IV.

## 2. Data and Methodology

### 2.1 Lightning Location System

The European Cooperation for Lightning Detection (EUCLID) operates an extensive network with over 170 sensors distributed throughout Europe. EUCLID's core mission is to identify cloud-to-ground (CG) strokes and intracloud (IC) pulses within the very low frequency/low frequency spectrum. The location of the electromagnetic signals is accomplished employing time of arrival (TOA) and magnetic direction finding (MDF) techniques. For every lightning strike that EUCLID identifies, it meticulously logs a comprehensive dataset. This includes an accurate timestamp to the sub-microsecond, the strike's

geographical coordinates, the type of the event (CG versus IC), the discharge's polarity, an estimate of the peak current, and detailed waveshape metrics such as risetime and the duration from peak to zero. Additionally, EUCLID records both direct

and inferred quality metrics, encompassing the semi-major and semi-minor axes of the 50% confidence ellipse for the event's location, the count of sensors that contributed to detecting the event, and the Chi$^2$ value, which assesses the agreement level among the participating sensors. The current configuration of the network is illustrated in Figure 1. For further details on EUCLID and its functionalities, please visit https://www.euclid.org.

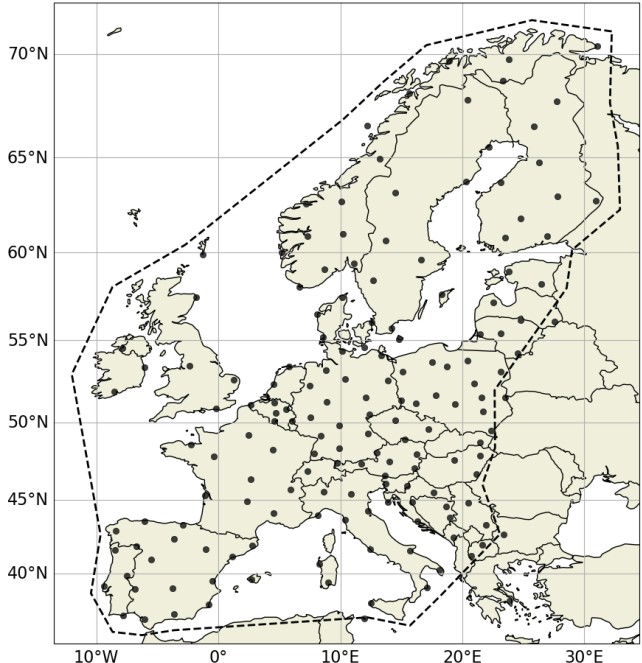

**Figure 1: Sensor locations within the EUCLID network. Only data within the dashed polygon is considered for quantitative analysis.**

The EUCLID network is dynamically evolving, consistently upgrading from older sensor models to newer ones and optimizing sensor placement by adding or relocating sensors. These modifications have been ongoing throughout the past and the period under investigation in this study from 2013 to 2022. Nevertheless, these adjustments are not substantial enough to significantly influence the results detailed in subsequent sections. This conclusion is supported by looking at the continuous performance evaluations with focus on the detection efficiency (DE) and location accuracy (LA) as outlined below.

EUCLID's evaluation strategy includes direct measurements from instrumented towers and additional data from video and electric field (E-field) recordings across various sites. Analysis of data from instrumented towers between 2007 and 2014 indicates an LA of approximately 100 meters (Schulz et al., 2016). Despite network changes, LA has shown steady improvement during that period. Regarding detection efficiency, the DE for negative CG strokes and flashes stands at 70% and 96%, respectively, based on tower data. However, when looking at video and E-field records from periods between 2008-2012, 2011, and 2012-2013, the stroke and flash DE is reported at 84% and 98%, respectively, as per findings by Schulz et al. (2016) and Diendorfer (2016). The DE derived from tower measurements typically underestimates that from natural downward

lightning because tower data primarily capture subsequent strokes, which generally exhibit lower peak currents and are more
challenging to detect. More recent evaluations by Schwalt et al. (2020) further confirm these findings, indicating stroke DE
rates between 76% and 85.6% from ground-truth data in Austria during 2015, 2017, and 2018. The consistency in DE metrics
across different studies and over time, despite the network's evolution, underscores the robustness and reliability of EUCLID's
lightning detection capabilities.

In addition to DE and LA, it is crucial to consider misclassification issues. Zhu et al. (2016) investigated the classification
accuracy (CA) of CG and IC lightning within the U.S. National Lightning Detection Network (NLDN), by comparing it to
optical and electrical field observations from the Lightning Observatory in Gainesville (LOG), Florida. The study found that
the NLDN achieved a CG stroke CA of 92%. For a total of 153 IC events, which included isolated IC events, IC events before
the first return stroke, and IC events after the first return stroke, the overall CA was 86%. Notably, the CA for isolated IC
events alone — representing complete IC flashes — was higher at 95%. This latter evaluation is pertinent to the EUCLID
network, as both systems utilize similar technology in terms of hardware and software, suggesting that the findings from NLDN
can provide insights applicable to EUCLID's operational context.

 It is assumed that the latter DE and LA values mentioned are applicable within the dashed polygon indicated in Fig. 1 and
diminishes as you move farther away from the network's periphery. Hence, for quantitative analysis, data within this polygon,
over the ten-year period from 2013 until 2022, is exclusively considered.

## 2.2 Ground strike point algorithm

Prior to attempting the merging of CG strokes into GSPs, it is necessary to group first individual strokes that are part of the
same lightning flash. In this process, which utilizes CG observations obtained from a LLS, the creation of artificial flashes
relies on well-defined and frequently employed spatial and temporal criteria. The criteria adopted here align with those
specified in the current IEC 62858 standard, i.e., strokes are considered part of a single flash if their time difference ($\Delta t$) is less
than 1.5 s and their spatial separation ($\Delta d$) is less than 10 km, both calculated in reference to the time and position of the first
stroke in the flash. Additionally, a temporal interstroke interval criterion of 0.5 s is applied. It is acknowledged that alternative
flash grouping algorithms exist, employing slightly different spatial and temporal settings. This consideration stems for
instance from research by San Segundo et al. (2020), who reported that 5-10% of lightning flashes might include multiple
GSPs spaced more than 10 km apart, and that it is rare for flashes to last longer than 0.8 seconds based on Lightning Mapping
Array (LMA) data. However, adjusting the maximum duration to 0.7 seconds and the grouping radius to 12 km would only
slightly increase the count of identified flashes by 0.5% (San Segundo et al., 2020). On the other hand, one has to recognize
that this current study also takes into account various additional factors impacting flash grouping, such as misclassification
from in-cloud to cloud-to-ground and detection efficiency effects, which have a more substantial effect than the modest 0.5%
change proposed by San Segundo et al. (2020). Consequently, the flash grouping algorithm utilized in this study is deemed
appropriate.

After allocating each individual stroke to their respective flashes, a GSP algorithm can be executed to group the observed CG strokes of a flash in one or more GSPs. What follows is a concise overview of the GSP algorithm utilized in this study. For a more comprehensive analysis of the algorithm's performance, readers who are interested are directed to (Poelman et al., 2021b; Poelman et al. 2023), where the algorithm is denoted as A1. The choice to use algorithm A1 is due to its use in processing the real-time data stream from EUCLID, making it the most representative method for capturing observations from EUCLID in this context. However, note that each algorithm described in Poelman et al. (2021b) has been shown to be highly accurate, suggesting that using a different algorithm would likely not significantly alter the findings of this study.

The GSP algorithm employs an iterative K-means approach by sequentially examining the strokes within a flash in chronological order. The initial position of the first GSP is determined by the location of the first stroke. Subsequently, the algorithm assesses the distance between each successive stroke and the existing GSPs against a predefined threshold value; being 500 m. If this distance is less than the threshold, the stroke is assigned to the nearest GSP. However, if the distance exceeds the threshold, a new GSP is created, provided that all distances to the existing GSP locations surpass the threshold. The positions of the GSPs are updated at the end of each iteration, with each stroke being assigned a weight that is inversely proportional to its respective semi-major axis (SMA) of the error ellipse. In addition, a stroke is assigned to the previous GSP regardless its position when the absolute peak current $|I_\mathrm{p}|$ is below 6 kA and/or the SMA is larger than 2 km. The latter is carried out to prevent the formation of spurious GSPs caused by imprecisely located strokes.

The selection of a 500-meter threshold is based on several factors. Firstly, at this threshold, the GSP algorithm demonstrates its optimal performance in distinguishing stroke types, as described in Poelman et al. (2021b). Secondly, when using a 200-meter threshold, the ratio of GSPs detected by the algorithm to the GSPs observed in high-speed video recordings (as detailed in Poelman et al. 2023) was approximately 1, but at the 500-meter threshold, it only underestimates the observed GSP count by no more than 5%, based on a limited ground-truth data set taken in France, Austria and Spain. Thirdly, notwithstanding the fact that the LA based on ground-truth data is significantly smaller than 500 m, it is important to note that ground-truth data are typically obtained from (and for) a limited number of sources (and locations), including a comparatively small set of tower measurements and some mobile video and E-field measurements. In turn, the LA can differ from one geographic location to the other, which may not be statistically reflected in the ground-truth data set. Conversely, when examining the semi-major axis of the 50% confidence error ellipse across Europe from 2013 to 2022, it's evident that SMA in central Europe consistently remains below 250 meters. Yet, as one moves to the periphery of the network, the SMA approaches 500 meters. Consequently, in order to adopt a standard applicable across the entire EUCLID domain, a 500-meter threshold has been selected to be on the safe side.

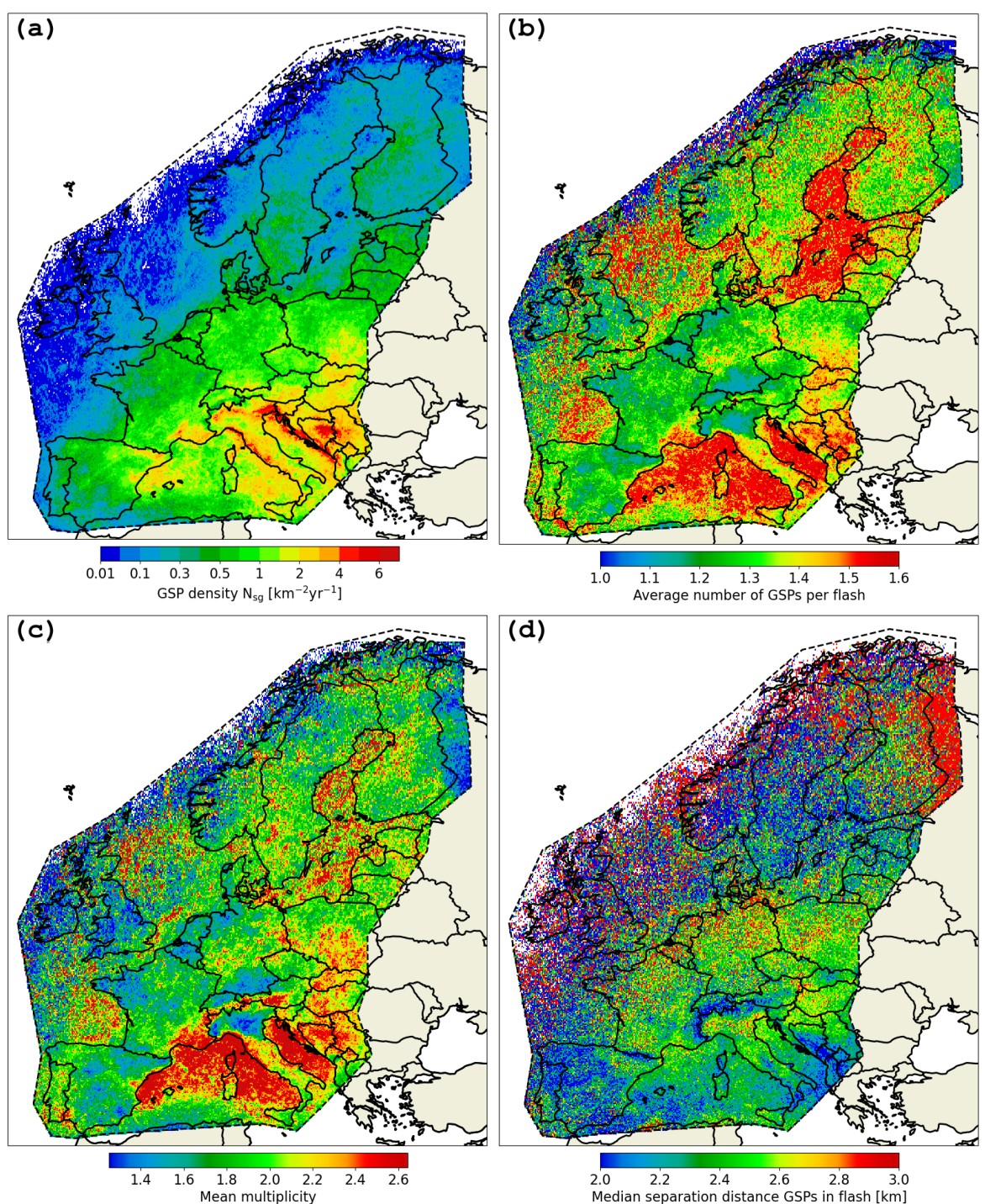

**Figure 2: Spatial distribution of the (a) mean annual ground strike point density $N_{SG}$ [km$^{-2}$ yr$^{-1}$], (b) average number of GSPs per flash (GSPF), (c) mean multiplicity, and (d) median separation distance amongst GSPs in a flash.**

## 3. Results

### 3.1 Spatial characteristics

In this section, we begin by examining observations made across the EUCLID domain. Following that, a specific area, intriguing due to its geographical location and deserving a more comprehensive investigation based on the upcoming results, is highlighted and explored in greater detail. The geographic plots that follow are displayed using a spatial resolution of 0.1° x 0.1°. This grid size is adequate to meet the minimum requirements outlined in Diendorfer (2008), ensuring a spatial uncertainty of less than 20% at a 90% confidence level throughout the EUCLID domain.

Figure 2a illustrates the average GSP densities, $N_{SG}$, across the European continent. It is expected that the observable variations in GSP densities correspond to the flash patterns outlined in a previous study conducted by Poelman et al. (2016b), meaning that, as a general trend, regions with higher flash densities tend to exhibit higher GSP densities. The highest concentrations are located near the Adriatic Sea and the western Balkan region, where they can reach a maximum of 8.5 GSPs per square kilometre per year.

When analysing the spatial pattern of the average number of GSPs per lightning flash (GSPF), as depicted in Figure 2b, a clear increase is observed in the Mediterranean, Adriatic, and Baltic Sea regions in contrast to inland regions. The investigation reveals that in the sea regions mentioned earlier, the average count of GSPF is approximately 1.5 or greater, while over land, this value decreases to around 1.3. However, there is a simple factor influencing this situation, which can be easily understood by examining Figure 2c. This figure provides a visual representation of the average multiplicity, i.e., the count of CG strokes within a single flash. Across the entire EUCLID domain, negative flashes exhibit an average multiplicity of 2.1, with a substantial 95% of the flashes comprising six strokes or fewer. The majority of negative CG flashes are single-stroke occurrences, making up 58% of the recorded flashes over the EUCLID domain. However, it's crucial to emphasize that the percentage of single stroke flashes represent an overestimation compared to values derived from ground truth recordings. It is closely associated with the stroke DE, the algorithm employed for grouping strokes into flashes, and, last but not least, the misclassification of IC pulses as CG strokes. It becomes evident that regions with higher multiplicity align with areas that exhibit a higher number of ground strike points per lightning flash. In essence, the figures 2b and 2c complement each other in demonstrating this relationship.

By utilizing the GSP positions within a flash, obtained through the GSP algorithm, it becomes feasible to calculate the distances between the various GSPs relative to all of them within that same flash. Figure 2d displays the spatial pattern of the median separation distance of GSPs in flashes. The median value is reported at 2.3 km, with the 10[th] and 90[th] percentiles measuring 900 meters and 5.5 km, respectively. A notable observation is the closer proximity of GSPs in a flash in the Alpine, Pyrenean, and Apennine regions. The connection between the distance between GSPs and the overall elevation of the terrain has also been previously noted by Cummins (2014). Cummins conducted a study on the influence of terrain on lightning occurrence in two distinct areas within the United States and conclusively established a spatial correlation between the absolute elevation of the terrain and the separation distance, particularly in the vicinity of the Rocky Mountains. This phenomenon appears to have

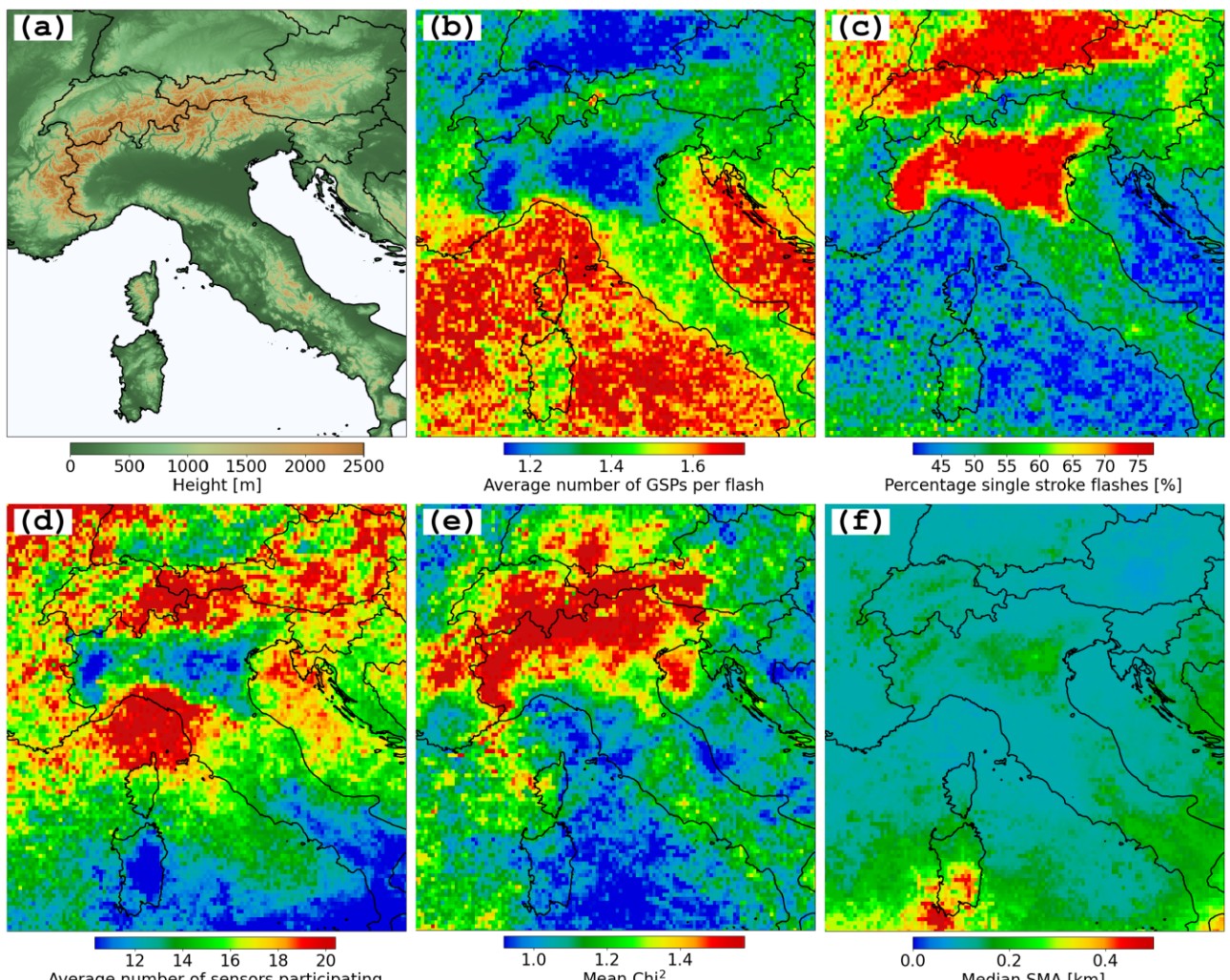

**Figure 3: (a) topography, (b) average number of GSPs per flash, (c) percentage of single stroke flashes, (d) average number of sensors participating, (e) mean Chi², and (d) median length of the semi-major axis of the 50% error ellipse.**

a valid physical explanation. Assuming a consistent height for preliminary breakdown and considering that lightning leaders often exhibit some degree of inclination as they move from the cloud toward the ground, it follows that in mountainous regions with higher altitudes, this would result in shorter channels and, consequently, shorter separation distances.

In the next part of this subsection, we focus on a particular geographical area that encompasses the Alps, Italy (including its islands), the western Balkan region, as well as portions of the Adriatic and Mediterranean Seas. The choice of this particular

region is twofold: firstly, its diverse topography, and secondly, equally significant, it is suitable for exploring the potential differences that might be present between land and sea. To achieve this objective, Figures 3 to 5 are utilized to evaluate the

potential influence of LLS performance on the outcomes and to conduct a more comprehensive examination of the observed GSP behaviour.

Figure 3a provides a representation of the topographical features in this region, showcasing the variations in altitude, notably the pronounced elevation of the Alps, and to a lesser extent, the Apennines, in comparison to their lower surrounding areas. Figure 3b displays an identical plot to that of Figure 2b, illustrating the average GSPF. It clearly highlights the contrast between land and sea, evident not only along the eastern and western coasts of Italy but also becoming apparent in the vicinity of Sardinia. The other sub figures of Fig.3 are used to delve deeper into exploring the factors that may account for the observed disparities in the average GSPF. Noticeable differences in the proportion of single-stroke lightning flashes are apparent when comparing the regions both north and south of the Alps, as well as the Alpine region itself, as depicted in Figure 3c. Furthermore, the percentage of single-stroke flashes is at its lowest over the sea in contrast to inland areas. The geographic distribution of single-stroke lightning flashes directly affects the distribution of GSPs per flash, particularly evident in the northern and southern regions of the Alps (see Fig. 3b). A similar connection is observed between the Mediterranean Sea and mainland Italy. The elevated occurrence of single-stroke flashes in the areas located to the north and south of the Alps can be attributed to the topographic influence of the Alps, which attenuates ground waves and consequently obstructs the sensors' ability to participate in locating lightning discharges. This phenomenon is primarily noted in the southern regions of the Alps, as depicted in Figure 3d where the average number of sensors participating is shown. In this specific area, the average participation of sensors in a solution decreases to approximately 10 to 12 sensors. A reduction is also observed to the north of the Alps, albeit to a lesser extent than in the areas immediately to the south of the Alps. It is worth observing that in the southern regions of Italy and Sardinia, the plot depicts the lowest average participation of sensors, which is associated with a high number of sensor outages during the investigated time period. However, this discrepancy does not result in a reduced proportion of single-stroke flashes when compared to what is witnessed in the northern and southern Alpine regions. Chi-square ($Chi^2$) values serve as an indicator of the consensus among the sensors employed for geolocating lightning discharges. In the case of a perfectly calibrated network, the average $Chi^2$ value is 1, while values below three are still considered satisfactory. Elevated $Chi^2$ values, on the other hand, are typically a result of larger arrival-time errors, which can be attributed to propagation effects. Observing Figure 3e, one can note that the most elevated $Chi^2$ values are situated in the Alpine region, peaking at 1.5. Beyond this particular region, the average $Chi^2$ values are at 1.2 or lower. This suggests that, overall, the network is well-calibrated, which is especially noteworthy considering the challenging topography in the area. Furthermore, the accuracy of error ellipse parameters, particularly the semi-major axis of the 50th percentile confidence ellipse, provides additional information about the network's quality. While there are instances where median SMA values are slightly greater than in other areas, Figure 3f illustrates that, in most cases, the median SMA value remains well below approximately 250 meters. In summary, the collective data from Figures 3d-3f strongly suggest that the EUCLID network is effectively calibrated. Investigating the underlying reasons for the high occurrence of single stroke flashes, though intriguing, falls beyond the scope of this study. However, the reduced number of participating sensors suggests that, likely due to lower conductivity in the Alps, signals may be attenuated or distorted on the opposite side of the Alps. Therefore, this could result in lightning sensors rejecting

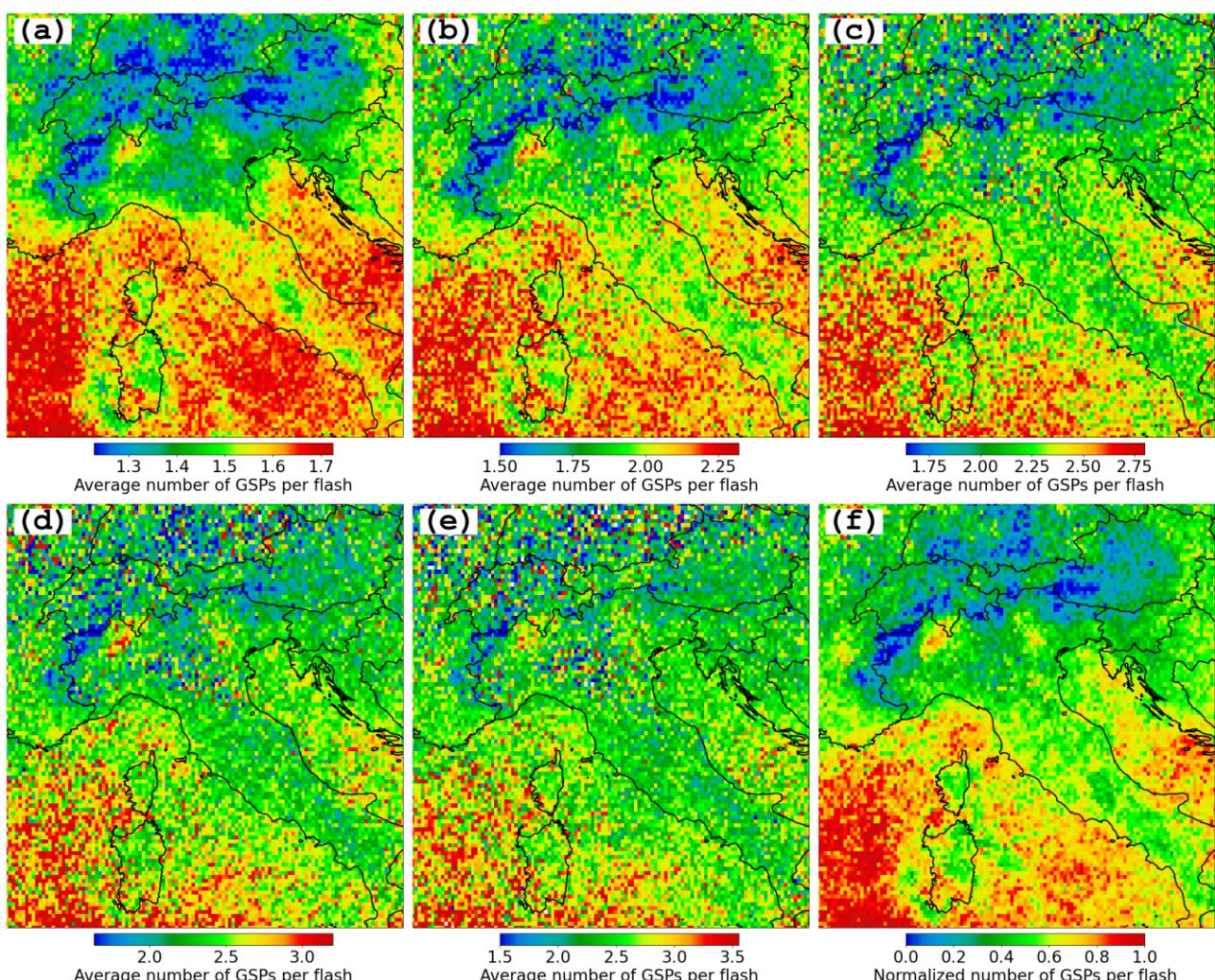

**Figure 4: The mean count of ground strike points per flash for flashes having a fixed multiplicity of (a) two, (b) three, (c) four, (d) five, and (e) six. A normalized spatial distribution of these mean counts for flashes with a constant multiplicity is presented in (f).**

the information either because of a lack of correlation with other sensors or when the signals are attenuated to a level below the sensor threshold.

As previously demonstrated in Fig. 3b and Fig. 3c, the multiplicity distribution significantly influences the spatial pattern of
the mean number of ground strike points per lightning flash. To mitigate the impact of the multiplicity distribution on the average GSPF and to genuinely explore whether flashes behave differently in terms of the production of new termination points across various topographic and conducting regions, the following method is adopted. Spatial plots are generated to illustrate the average GSPF, exclusively for flashes with particular multiplicities. This methodology enables the investigation

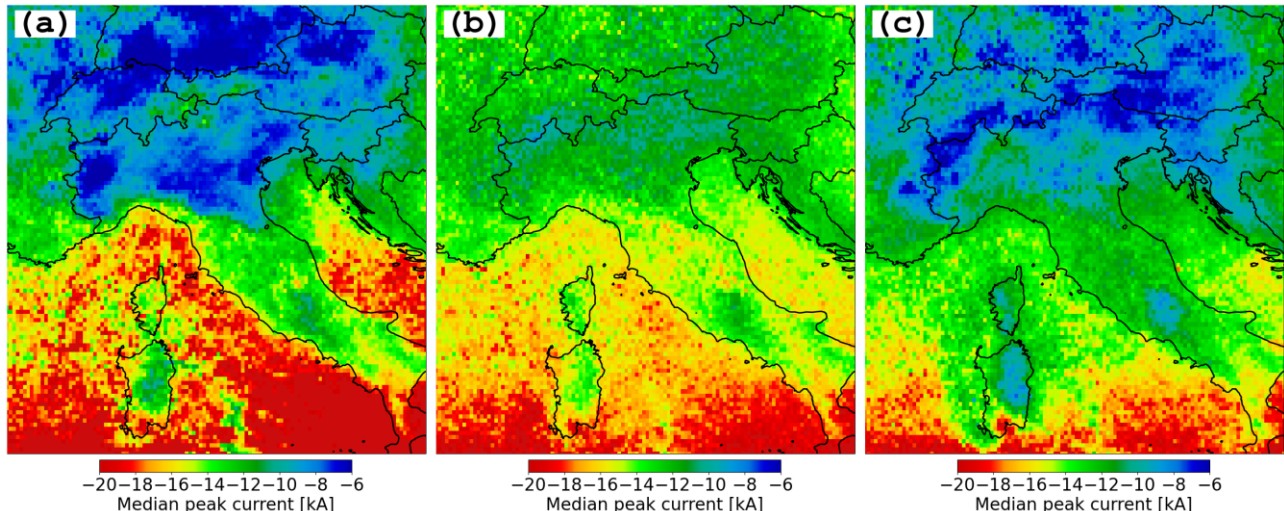

**Figure 5: Spatial distribution of the median peak current amplitude for (a) flashes, (b) ground strike points excluding first strokes in a flash, and (c) subsequent strokes in ground strike points.**

of potential fluctuations in the average GSPF. Fig. 4 illustrates this process, presenting the mean count of ground strike points per lightning flash for flashes with fixed multiplicities ranging from 2 to 6. Two noteworthy observations emerge from this analysis. First, the average GSPF for flashes with fixed multiplicity is at its lowest over the Alps. Second, a distinct land/sea disparity is evident. To alleviate the influence of multiplicity while considering flashes of all multiplicities, each of Fig. 4a–4e is scaled to values between 0 and 1. Subsequently, the spatial plots are aggregated by summing the values of the corresponding grid cells. The resulting composite outcome is once again normalized to ensure a scale ranging from 0 to 1. The normalized output seen in Fig. 4f provides a more accurate representation of how GSPs behave within a flash across the region.

The reason behind the observations in Fig. 4, in particular Fig. 4f, raises questions. An unexplored path relates to the analysis of the peak current distribution and its potential influence on the formation of new ground termination points. In previous studies it has been demonstrated that on average peak currents over sea tend to be stronger compared to the peak currents over land (Cooray et al., 2014, Nag and Cummins, 2017 Poelman et al., 2016). Cooray et al. (2014) was, to the best of the author's knowledge, the first to suggest that variations in meteorological conditions during storms underlie the distinctions observed between maritime thunderclouds and those occurring over land. The conclusion drawn was that the lack or reduction of the essential prerequisites for the development of positive charge pockets below the main negative charge center within maritime clouds results in ground lightning flashes originating at a higher electrical potential within the negative charge area, in contrast to those initiated in land-based storms. Additionally, Nag and Cummins (2017 discovered that the median duration of the stepped-leader, which is the time gap between preliminary breakdown pulses and the negative first stroke, is briefer when observed above the ocean compared to over land. They provided evidence that these shorter durations over the ocean correspond to the higher oceanic peak current levels that have been documented. Fig. 5 illustrates the spatial distribution of

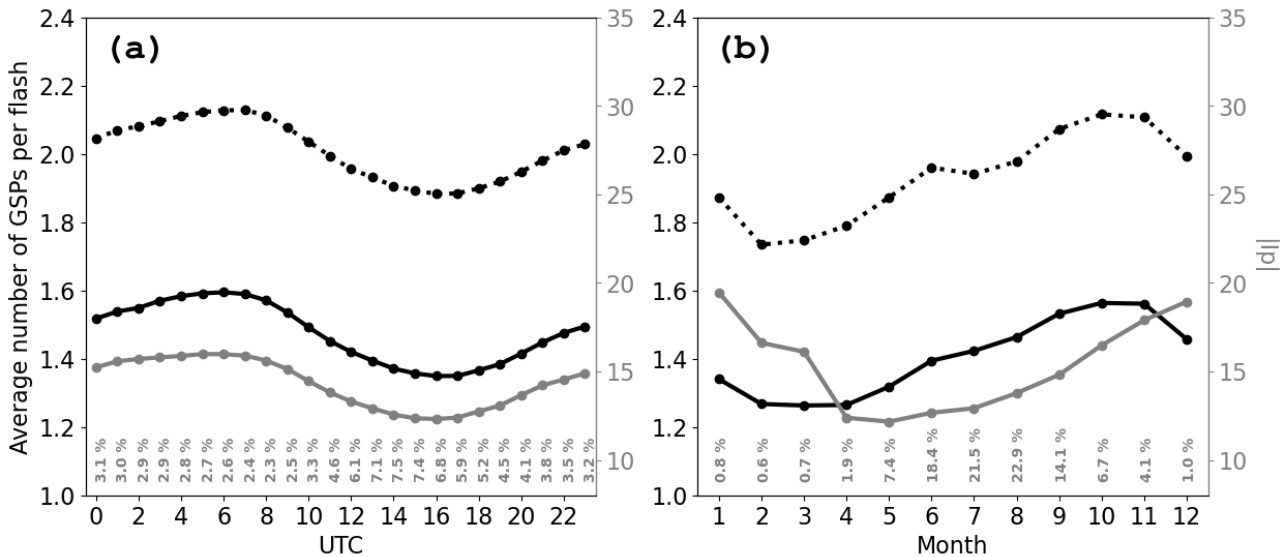

**Figure 6: (a) Diurnal and (b) seasonal variation of the average number of GSPs per flash, for all flashes (solid black) and excluding single stroke flashes (dotted black). The median value of the absolute peak current is represented in grey. The numbers at the bottom indicate the proportion of activity, where 1% corresponds to approximately 700,000 strokes.**

median peak current for three categories: flashes, GSPs (excluding the first strokes within a flash, which also entails the exclusion of single-stroke flashes), and the subsequent strokes within GSPs, whereby the peak current of a GSP is defined as the peak current of the initial stroke linked to that particular GSP. The flash peak currents exhibit a noticeable decrease to the north and south of the Alps. This minimum arises from the high occurrence of single-stroke flashes in those regions, as we will later demonstrate that flash peak current increases with multiplicity. Consequently, this leads to the observed minimum.

The median peak current for GSPs — excluding the first strokes within flashes — is at its lowest over the Alps, as is the case for the median values of subsequent strokes in GSPs. Per definition, the latter two categories of lightning strokes potentially contribute to the formation of new termination points or may opt to trace an existing channel. In both scenarios, the least average GSPF in the Alps coincides with the lowest observed peak current. An additional strengthening factor that could also come into play in the Alps or any mountainous terrain is that sharp peaks tend to draw lightning discharges more easily.

Consequently, within a lightning flash, the probability of the discharge being drawn to the same peak, rather than establishing a new termination point at a greater distance, is enhanced. The same observation applies at the instrumented Gaisberg tower, where the multiplicity of flashes is generally higher compared to what is observed in natural cloud-to-ground lightning. Apart from the distinctions observed in the Alps, it's evident that a noticeable contrast exists between land and sea. Our findings in this study align with earlier research in that the median peak current for flashes is greater over the sea compared to over land.

What's particularly noteworthy and not previously documented is that this pattern also extends to the peak current of GSPs and the subsequent strokes within GSPs as can be seen in Fig. 5b and Fig. 5c.

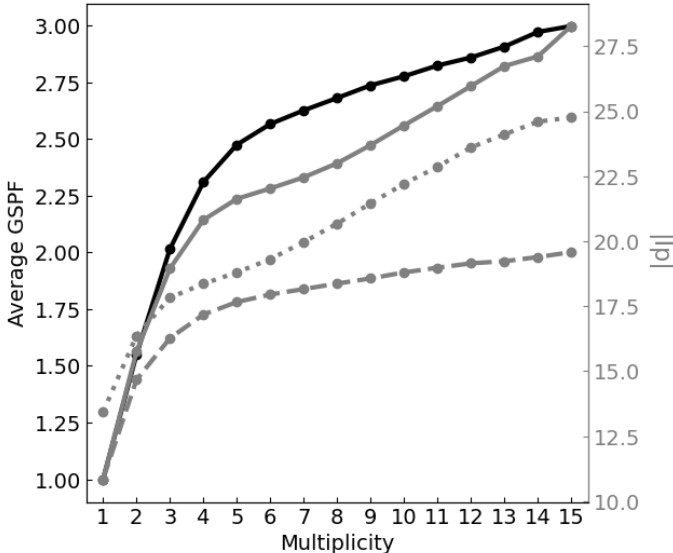

**Figure 7: The graph illustrates the average number of GSPs per flash (GSPF) in relation to the flash multiplicity, depicted in black. The median absolute peak current values for flashes (solid) and for GSPs within flashes (dashed) are presented in grey, also in relation to the flash multiplicity. Additionally, the absolute median 1st stroke peak current of GSPs as a function of GSP multiplicity, i.e., the number of strokes allocated to a specific GSP, is shown (dotted).**

Additionally, this transition from land to sea conditions is evident around the regions of Sardinia and Corsica. While the conditions of the propagation path have been shown to have an impact on various lightning parameters measured by LLS networks (see for example Chisholm, 2017), the above results support the idea that atmospheric distinctions within thunderclouds play a primary role in the observed land-sea transition of lightning peak currents. Also in this case it is observed that maritime regions exhibiting higher peak current amplitudes have a higher average GSPF. The latter implies that the observed distributions of peak current and GSPF share a common root cause. According to Nag and Cummins (2017), clouds electrified over the ocean may exhibit a larger main negative charge region and a smaller lower positive charge region. This in turn could lead to negative leaders with elevated leader tip potential and a higher line charge density, resulting in faster average vertical velocity during downward propagation. Consequently, it is probable that increased leader tip potential not only leads to elevated peak currents but also concurrently raises the likelihood of branching-off instead of following a pre-existing ionized channel.

### 3.2 Temporal characteristics

Figure 6 depicts the fluctuations in both daily and seasonal average GSPF, alongside the peak values. As shown in Figure 6a, the highest peak current is observed during the evening to early morning hours (approximately 22-8 UTC), while it is at its lowest in the afternoon. Additionally, the peak in CG lightning activity aligns with the minimum in absolute peak current.

Chronis et al. (2014) previously identified this pattern and suggested that the frequent occurrence of CG lightning in convectively active conditions might hinder the accumulation of electrostatic charge within thunderclouds, leading to less intense discharges. What is intriguing is that the distribution of the average GSPF closely mirrors the distribution of peak currents. This once more indicates that the same principles discussed in the concluding part of Sect. 3.1 are applicable in this context as well. A comparable trend can be observed in Figure 6b, but now at a monthly interval. The lowest peak current values are typically observed during the summer months, with higher values occurring at the transition between the end and start of the year. This phenomenon has been documented previously (Brook, 1992) and is thought to be linked to the observation that electric field initiation of lightning is more pronounced during winter compared to summer. Consequently, discharges occurring in winter tend to be more energetic, featuring higher peak currents. The relationship between the average GSPF and the absolute peak current is noticeable on a monthly scale, but it is not as strong as it is at the daily level.

## 3.3 Additional characteristics

Figure 7 presents the average GSPF in relation to the flash multiplicity. Furthermore, it features the median absolute peak current values both for flashes and for GSPs within a flash as a function of the flash multiplicity. Additionally, it illustrates the absolute median peak current of GSPs as a function of the GSP multiplicity, i.e., the number of strokes associated to a GSP. Note that the peak current of a flash corresponds to the peak current of its initial stroke, and similarly, the peak current of a GSP is determined by the peak current of the first stroke associated with that GSP. Certainly, as the flash multiplicity increases, the likelihood of generating additional termination points also grows, as is demonstrated by the black curve. GSPF exhibits a steep increase at lower multiplicity levels, while the rate of increase is levelling off at flashes of even higher multiplicity. Regarding the median peak current value in single-stroke flashes versus higher multiplicity flashes, it is observed that the peak current of the initial stroke in the flash is approximately twice as high in flashes with a multiplicity of three compared to single-stroke flashes (solid grey). A similar upward trend in flash peak current concerning flash multiplicity is also identified for the GSP peak current with respect to GSP multiplicity (depicted in dotted grey). Finally, it is observed that the median peak current of the GSPs within a flash is higher in flashes with increased multiplicity (depicted in dashed grey).

Finally, Fig. 8a presents the median peak current values of GSPs, sorted by their order of occurrence within a lightning flash, while Fig. 8b illustrates the median peak current of strokes in relation to the order within the corresponding GSP. According to Figure 8a, it can be inferred that the median peak current value for the first GSP in a flash is -14.3 kA, which corresponds to the median value when aggregating all the peak currents from flashes of various multiplicities as depicted in Figure 7. Interestingly, the peak current of the second GSP in a flash is approximately 1 kA greater than the flash peak current. This has been seen as well when comparing the spatial plots in Fig. 5a and 5b. This could appear surprising, but remember that, for example, the stroke responsible for generating the second GSP is not necessarily the second stroke in the flash. As demonstrated

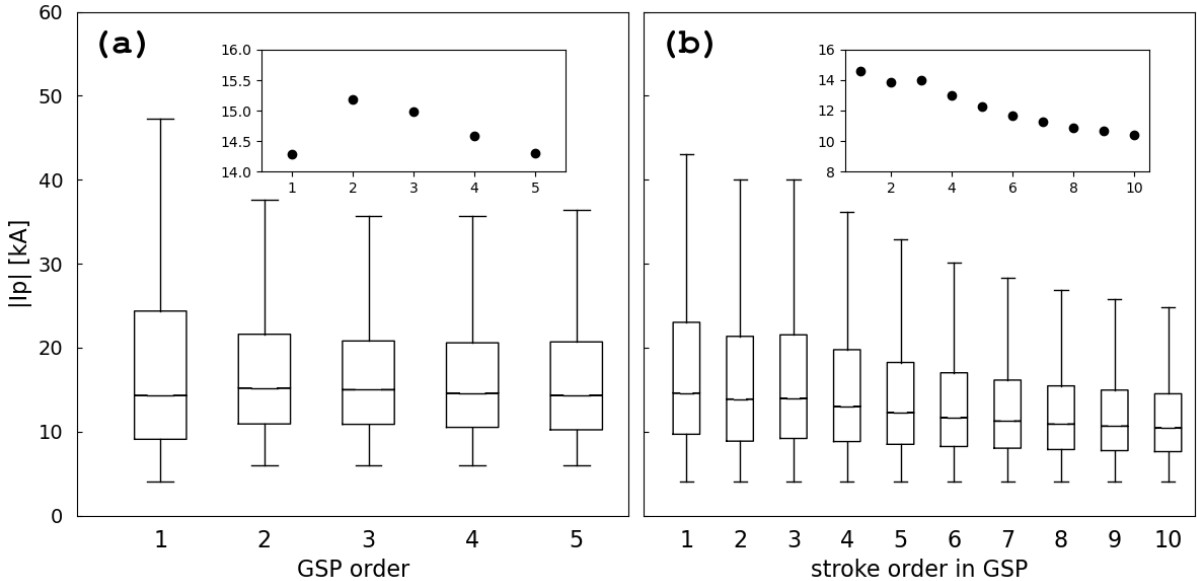

**Figure 8: (a) median absolute peak current of GSP in relation to its order of occurrence within the flash, and (b) median absolute peak current of strokes as a function of stroke order within the flash.**

in Figure 8b, the median peak current conforms to a more expected pattern. It shows that the first stroke within a GSP exhibits the highest peak current and gradually decreases for subsequent strokes within that GSP.

## 4. Summary and Conclusions

This study investigates the characteristics of ground strike points, observed through the ground-based LLS EUCLID. The investigation places emphasis on topographical impact, distinctions between land and sea areas, and how the network's performance affect the results. The spatial distribution of GSPs reveals elevated concentrations in proximity to the Adriatic Sea and the western Balkan region, closely correlating with the distribution of flash multiplicity. Nonetheless, when the influence of flash multiplicity is removed from the spatial pattern of GSPs, a distinct pattern emerges. It becomes evident that the average number of ground strike points per flash is greater over sea regions in contrast to inland areas, with a minimum observed in the Alps. Furthermore, the spatial distribution of the median separation distance among GSPs within a flash clearly highlights that this value is at its lowest in mountainous regions like the Alps and the Pyrenees, displaying the significant role of terrain elevation in this context. An effort was made to explore the link between the peak current of strokes and the formation of new termination points. The study revealed that, at a daily level, the peak current variation closely aligns with the distribution of GSPF, and this relationship persists at a monthly level, albeit with somewhat less strength. Additionally, it has been demonstrated that the peak current of GSPs, excluding the first strokes in a flash, as well as the subsequent strokes within GSPs, is lower over the Alpine regions and highest over the sea (see Fig. 5b and 5c), exactly corresponding to the observed

pattern in GSPF. We argue that the observed distributions of peak current and GSPF stem from a common underlying cause. Assuming the validity of the hypothesis proposed by Nag and Cummins (2017), the lack or reduction of a positive charge pocket and larger main negative charge in maritime thunderclouds, results in a field configuration that leads to an enhanced leader tip potential. This sequence of effects culminates in increased peak currents and an increased likelihood of establishing a new termination point. However, the mechanism explaining how or why increased peak current leads to a higher probability for the subsequent leader to branch off remains a topic for further investigation.

Within the domain of lightning protection and risk calculation, the selection of an appropriate multiplier for ground impact points per CG flash has long been a subject of discussion and was prompted at the time since LLSs only reported flash densities. Initially, Bouquegneau et al. (2012) hinted at the necessity of applying a robust safety factor in risk component calculations, which could involve adjusting the value of $N_G$. Building upon this, Rousseau et al. (2019) further reconfirmed doubling the $N_G$ value in cases where $N_{SG}$ is not obtained from a lightning detection system that meets the IEC 62858 standards, established by the International Electrotechnical Commission (2019). This approach aims to ensure a sufficient safety margin in risk assessments. On the other hand, the CIGRE TB 549 report by the International Council on Large Electric Systems, released in 2013, suggests a more modest correction factor between 1.5 and 1.7, when only flash density data are accessible. One way or another, the optimal method involves directly calculating strike point density using a comprehensive lightning location network according to IEC 62858, made possible with present day state-of-the-art LLSs.

Recent research, such as the study by Vagasky et al. (2024) along with the results of this study, suggests that doubling $N_G$ may significantly overestimate actual needs. This is supported by our findings indicating that most regions within the EUCLID domain have a ratio of less than 1.6 ground strike points per CG flash (see Fig. 2b). Therefore, although using a factor of two to estimate $N_{SG}$ offers a method to enhance lightning protection when only $N_G$ data are accessible, it may overestimate the risk.

**Data availability.**

The EUCLID data used in this study can be provided upon request.

**Author contributions.**

DRP analysed the EUCLID dataset, while HK and WS provided feedback at every step in the process. DRP prepared the manuscript with review and editing from HK and WS.

**Competing interests**

The contact author has declared that none of the authors has any competing interests.

**Acknowledgments.**

The authors are grateful for the input provided by the reviewers. Their thoughtful comments have substantially enhanced both the depth and presentation of the results. We sincerely appreciate the time and effort they have dedicated to this work.

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
