# Peer review of "Insights into ground strike point properties in Europe through the EUCLID Lightning Location System"

_EGUsphere, 2024_

## Author Comment (AC1)

**Referee #1: Hunt, Hugh  hugh.hunt@wits.ac.za**
**nominated 06 Feb 2024, accepted 13 Feb 2024, report 13 Feb 2024**

This is a very interesting and relevant study and yields important contributions to our understanding of lightning (and in particular, downward lightning behaviour).

The paper is well-organized, with a clear structure that follows the conventional format of academic papers. The use of figures to illustrate spatial distributions and properties of GSPs enhances the clarity and understanding of the research findings. However, a more detailed discussion on the limitations of the study and potential areas for future research could further strengthen the paper

I have a couple of comments for discussion:

1. I am interested in the choice of 500 m as the distance threshold used. As noted, the EUCLID network consistently is found to have a location accuracy below 250 m and, while there is not a significant increase in the accuracy of the algorithms, would it not be worth using to achieve more accurate GSP classification?

   The reviewer's observation that the location accuracy (LA) based on ground-truth data falls significantly short of 500 meters is accurate. However, it's important to note that such ground-truth data are typically derived from a limited number of sources, including a handful of tower measurements and some mobile video and E-field measurements. Despite considerable efforts to compile these data sets, their major limitation is their sparse geographic coverage. Conversely, when examining the semi-major axis of the 50% confidence error ellipse (SMA) across Europe from 2013 to 2022, it's evident that SMA in central Europe consistently remains below 250 meters. Yet, as one moves to the periphery of the network, the SMA approaches 500 meters. Consequently, in order to adopt a standard applicable across the entire EUCLID domain, a 500-meter threshold has been selected. Moreover, previous research (Poelman et al. 2023)[1] supports this decision as pointed out by the reviewer, demonstrating that the discrepancy in underestimating or overestimating the number of Ground Strike Points (GSPs) with a threshold of 250 meters versus 500 meters is minimal, as also noted in the document.

   On the other hand, we do acknowledge that in the case of specific local lightning protection studies, e.g., for a private company, it might be worthwhile to examine in a bit more detail the potential changes that may occur when adjusting the thresholds.

[1]Poelman, D. R., H. Kohlmann, W. Schulz, S. Pedeboy and L. Schwalt, "Ground strike point properties derived from observations of the European Lightning Location System EUCLID," 2023 12th Asia-Pacific International Conference on Lightning (APL), Langkawi, Malaysia, 2023, pp. 1-5, doi: 10.1109/APL57308.2023.10182055

Similarly, while there is not a large accuracy increase between algorithms, why was the algorithm that does not take into account ellipse information chosen over the other algorithms?

The reviewer points out correctly that each algorithm detailed in Poelman et al. (2021)[2] demonstrates high accuracy, indicating that the choice of algorithm would not significantly change the results presented in this study. The real-time data stream from EUCLID employs algorithm A1, which has thus been selected as the most representative method for capturing EUCLID observations in this study as well.

[2]Poelman, D. R., Schulz, W., Pedeboy, S., Campos, L. Z. S., Matsui, M., Hill, D., Saba, M., Hunt, H.: Global ground strike point characteristics in negative downward lightning flashes – part 2: Algorithm validation, *Nat. Hazards Earth Syst. Sci.*, 21, 1921-1933, **2021,** https://doi.org/10.5194/nhess-21-1921-2021

1. Flash density is a key parameter for assessing lightning risk in lightning protection design and GSP density has an important implication for lightning risks. Recent recommendations to multiply Ng by a factor of 2 to approximate Nsg and include this in the IEC62305 approaches make this study particularly relevant.

It would be good to see some discussion of this in the manuscript. In particular, comment on whether a factor of 2 is appropriate?

Very good suggestion by the reviewer. We propose to add following to the manuscript:

Within the domain of lightning protection and risk calculation, the selection of an appropriate multiplier for ground impact points per CG flash has long been a subject of discussion and was prompted at the time since LLSs only reported flash densities. Initially, Bouquegneau et al. (2012)[3] hinted at the necessity of applying a robust safety factor in risk component calculations, which could involve adjusting the value of Ng. Building upon this, Rousseau et al. (2019)[4] further reconfirmed doubling the Ng value in cases where Nsg is not obtained from a lightning detection system that meets the IEC 62858 standards, established by the International Electrotechnical Commission in 2019[5]. This approach aims to ensure a sufficient safety margin in risk assessments. On the other hand, the CIGRE TB 549 report[6] by the International Council on Large Electric Systems, released in 2013, suggests a more modest correction factor between 1.5 and 1.7, when only flash density data are accessible. One way or another, the optimal method involves directly calculating strike point density using a comprehensive lightning location network according to IEC 62858, made possible with present day state-of-the-art LLSs.

Recent research, such as the study by Vagasky et al. (2024)[7] along with the results of this study, suggests that doubling Ng may significantly overestimate actual needs. This is supported by our findings indicating that most regions within the EUCLID domain have a ratio of less than 1.6 ground strike points per CG flash (see Fig. 2b). Therefore, although using a factor of two to estimate Nsg offers a method to enhance lightning protection when only Ng data are accessible, it may also lead to unnecessary expense.

[3]Bouquegneau, C., A. Kern, and A. Rousseau, 2012: Flash density applied to lightning protection standards. Proc. GROUND 2012, Bonito, Brazil, Brazilian Society for Electrical Protection
[4]Rousseau, A. S., F. Cruz, S. Pedeboy, and S. Schmitt, 2019: Lightning risk: How to improve the calculation? Int. Colloquium on Lightning and POower Systems, Delft, Netherlands, CIGRE
[5]International Electrotechnical Commission, 2019: IEC 62858:2019: Lightning density based on lightning location systems – General principles
[6]International Council on Large Electric Systems, 2013: Lightning parameters for engineering applications. Working Group C4.407, CIGRE TB 549
[7]Vagasky, C., R. L. Holle, M. J. Murphy, J. A. Cramer, R. K. Said, M. Guthrie, and J. Hietanen, 2024: How Much Lightning Actually Strikes the United States? Bull. Amer. Meteor. Soc., 105, E749–E759, https://doi.org/10.1175/BAMS-D-22-0241.1.

1. The final sentence of Section 3.2 "The relationship between the average GSPF and the absolute peak current is noticeable on a monthly scale, but it is not as strong as it is at the daily level."

Is this correct? The relationship seems clear at a daily level…

The original sentence in the manuscript conveys that there is a moderate correlation between average Ground Strike Point per Flash (GSPF) and absolute peak current on a monthly basis, but this correlation is not _as_ strong _as_ the one observed on a daily scale. On a daily level, the trends between GSPF and absolute peak current align almost perfectly.

---

## Author Comment (AC2)

**nominated 08 Mar 2024, accepted 12 Mar 2024, report 01 Apr 2024**

The paper mainly discusses the ground strike points (GSP) spatial distribution over Europe based on cloud-to-ground (CG) stroke data from EUCLID network. It has a very short introduction that does not bring a clear motivation for the study. The description of the data and methods is fine, and the results are presented by nice figures and plots. However, the discussion is a little confusing, mixing different aspects of lightning detection that prevents from a good understanding.

1) My main concern is the actual motivation of the paper. Although the analyzes of the spatial distribution and the temporal characteristics of GSP are important for lightning protection, the paper does not discuss why we need of those results. In the Introduction (lines 26-32), it was said: "Among the various components that influence the risk estimation, the standard puts forward the flash density, NG, representing the number of lightning flashes per square kilometer per year, as one of the key parameters. However, by definition, the location of a flash is determined by the position of the first cloud-to-ground (CG) stroke within the flash. On the other hand, numerous studies, supported by high-speed camera observations (Rakov et al., 1994; Valine et al., 2002; Saraiva et al., 2010; Poelman et al., 2021a), have provided evidence that, on average, multiple ground strike points (GSPs) exist within multiple-stroke flashes. Hence, GSP densities should be given the pivotal role in lightning studies, particularly in the context of assessing lightning-related risks". The authors shall discuss more comprehensively how this study can effectively improve the lightning protection standards "in the context of assessing lightning-related risks". Remind that NHESS main scope is natural hazards, and the main topic of the paper must be connected to this subject.

This remark is in line with the remarks raised by reviewer 1 and reviewer 2. We propose to add following to the manuscript:

Within the domain of lightning protection and risk calculation, the selection of an appropriate multiplier for ground impact points per CG flash has long been a subject of discussion and was prompted at the time since LLSs only reported flash densities. Initially, Bouquegneau et al. (2012)[1] hinted at the necessity of applying a robust safety factor in risk component calculations, which could involve adjusting the value of Ng. Building upon this, Rousseau et al. (2019)[2] further reconfirmed doubling the Ng value in cases where Nsg is not obtained from a lightning detection system that meets the IEC 62858 standards, established by the International Electrotechnical Commission in 2019[3]. This approach aims to ensure a sufficient safety margin in risk assessments. On the other hand, the CIGRE TB 549 report[4] by the International Council on Large Electric Systems, released in 2013, suggests a more modest correction factor between 1.5 and 1.7, when only flash density data are accessible. One way or another, the optimal method involves directly calculating strike point density using a comprehensive lightning location network according to IEC 62858, made possible with present day state-of-the-art LLSs.
Recent research, such as the study by Vagasky et al. (2024)[5] along with the results of this study, suggests that doubling Ng may significantly overestimate actual needs. This is supported by our findings indicating that most regions within the EUCLID domain have a ratio of less than 1.6 ground strike points per CG flash (see Fig. 2b). Therefore, although using a factor of two to estimate Nsg offers a method to enhance lightning protection when only Ng data are accessible, it may also lead to unnecessary expense.

[1]Bouquegneau, C., A. Kern, and A. Rousseau, 2012: Flash density applied to lightning protection standards. Proc. GROUND 2012, Bonito, Brazil, Brazilian Society for Electrical Protection
[2]Rousseau, A. S., F. Cruz, S. Pedeboy, and S. Schmitt, 2019: Lightning risk: How to improve the calculation? Int. Colloquium on Lightning and POower Systems, Delft, Netherlands, CIGRE
[3]International Electrotechnical Commission, 2019: IEC 62858:2019: Lightning density based on lightning location systems – General principles
[4]International Council on Large Electric Systems, 2013: Lightning parameters for engineering applications. Working Group C4.407, CIGRE TB 549
[5]Vagasky, C., R. L. Holle, M. J. Murphy, J. A. Cramer, R. K. Said, M. Guthrie, and J. Hietanen, 2024: How Much Lightning Actually Strikes the United States? Bull. Amer. Meteor. Soc., 105, E749–E759, https://doi.org/10.1175/BAMS-D-22-0241.1.

2) In lines 147-180 there is a long discussion regarding the lightning location limitations on measuring the CG strokes. I'm not sure if all those details are necessary. Chi2 and SMA are quality solution parameters that are used to select "good" solutions, which were used in the analyzes. In my opinion, these parameters might only be relevant for the study if the authors describe more comprehensively how EUCLID detects and geolocates lightning. Even in this case, I was wondering if this discussion can be suppressed.

Fig. 2 illustrates spatial plots across the entire EUCLID domain. While it is out the scope of this study to go in detail about the performance of the network on many different spatial sub-domains, the authors believe it is both relevant and informative to delve into a more detailed analysis within the specifically zoomed-in region presented in Figures 3 to 5. This examination aims in particular to shed light on the intricacies observed in relation to GSPF. Our goal is to illustrate that the EUCLID network exhibits robust calibration, especially when dealing with the challenging topography. However, it is important to note that, even with the effective calibration, the topography can still influence the spatial distribution of GSPF. This "exercise" highlights the network's precision while acknowledging the complex interplay between topography and performance metrics.

The authors propose to adapt what was already present in Sect. 2.1 in order to meet the request of the reviewer 'how EUCLID detects and geolocates lightning' as follows: "The network's primary function is to identify cloud-to-ground strokes (CG) and intracloud (IC) pulses within the very low frequency/low frequency spectrum. The location of the electromagnetic signals is accomplished employing time of arrival (TOA) and magnetic direction finding (MDF) techniques. For every lightning strike that EUCLID identifies, it meticulously logs a comprehensive dataset. This includes an accurate timestamp to the sub-microsecond, the strike's geographical coordinates, the nature of the event (distinguishing between cloud-to-ground (CG) and intracloud (IC) discharges), the discharge's polarity, an estimate of the peak current, and detailed waveshape metrics such as risetime and the duration from peak to zero. Additionally, EUCLID records both direct and inferred quality metrics, encompassing the semi-major and semi-minor axes of the 50% confidence ellipse for the event's location, the count of sensors that contributed to detecting the event, and the $Chi^2$ value, which assesses the agreement level among the participating sensors."

3) The paper also discusses the land-ocean peak-current contrast, which has already been described by several other publications cited by the authors: Cooray et al., 2014, Nag and Cummins, 2017 Poelman et al., 2016. I do not find any contribution of the GSP analysis to this topic. The same for the diurnal and seasonal variations (Figure 6). Is the intention of the authors only to present the GSP temporal behaviors? If yes, then you are please encouraged to discuss in more details how these characteristics impact on lightning protection and even on natural hazards.

The distinction between land and ocean peak currents has indeed been explored in previous research mentioned by the reviewer, with a focus on the flash and stroke level, rather than on GSP level as depicted in Figures 5a and 5b. The authors consider this to represent a subtle, yet significant, nuance when compared to the existing literature.

While presenting the spatial and temporal patterns is one aspect (Figures 5 and 6), our study goes further by attempting to establish a plausible connection observed between GSPF and GSP peak currents. Specifically, we argue that the observed temporal and spatial distributions of peak currents and GSPF may originate from a shared underlying cause. The authors recognize that the current data available from the EUCLID network limits our ability to delve deeper or provide further evidence for the relationship previously mentioned. Our inclination is towards utilizing this information to determine whether targeted laboratory experiments could provide additional insights into the hypothesis. For example, it could be interesting to explore whether an increase in leader tip potential could increase the tendency for branching-off, rather than following an already ionized path.

The scope of NHESS includes the detection, monitoring of natural phenomena […] and the spatial and temporal evolution of hazardous natural events […]. Therefore, the authors consider the temporal characteristics of GSPs highlighted in this study to be pertinent to the topic outlined by NHESS. Note that in the past, the authors have published a similar paper in NHESS, i.e., Poelman et al. (2016) describing the spatial and temporal observations of CG flashes in EUCLID.

4) Finally, the correlations of the GSP with multiplicity and peak current are discussed based on Figures 7 and 8. Again I do not find any relevance of these results in terms of GSP analysis, lightning protection, or natural hazards. I'd like to see a more comprehensive discussion on how these parameters affects the GSP results which will consequently impact on the lightning protection standards.

The authors are of the opinion that the analysis showcased in Figures 6 and 7 is indeed relevant to this study. While the peak current analysis depicted in these figures may not have a direct impact on the implementation of effective lightning protection measures, it offers scientific insights that align with findings reported in the existing literature concerning the peak current of cloud-to-ground (CG) strokes relative to their occurrence within a CG flash.

---

## Author Comment (AC3)

**nominated 08 Mar 2024, accepted 08 Mar 2024, report 28 Mar 2024**

The paper is well-organized, well-written, and is an important contribution to the lightning detection community, especially in improving the lightning risk evaluation and related standards. Still, some aspects should be further discussed to emphasize the results.

Specific comments

The paper stresses that flash density (NG) is a crucial parameter to evaluate lightning risk (e.g. IEC 62305). However, there's no further mention of the norm neither in the discussion nor the conclusions. Results suggest the risk estimation of being struck by lightning should move to ground striking points (GSP) instead of NG. This point is relevant and should be discussed, as lightning risk has been introduced in the paper as one of the possible applications of the obtained results, but not further mentioned.

This remark is in line with the remark raised by reviewer 1. We propose to add following to the manuscript:

Within the domain of lightning protection and risk calculation, the selection of an appropriate multiplier for ground impact points per CG flash has long been a subject of discussion and was prompted at the time since LLSs only reported flash densities. Initially, Bouquegneau et al. (2012)[1] hinted at the necessity of applying a robust safety factor in risk component calculations, which could involve adjusting the value of Ng. Building upon this, Rousseau et al. (2019)[2] further reconfirmed doubling the Ng value in cases where Nsg is not obtained from a lightning detection system that meets the IEC 62858 standards, established by the International Electrotechnical Commission in 2019[3]. This approach aims to ensure a sufficient safety margin in risk assessments. On the other hand, the CIGRE TB 549 report[4] by the International Council on Large Electric Systems, released in 2013, suggests a more modest correction factor between 1.5 and 1.7, when only flash density data are accessible. One way or another, the optimal method involves directly calculating strike point density using a comprehensive lightning location network according to IEC 62858, made possible with present day state-of-the-art LLSs.

Recent research, such as the study by Vagasky et al. (2024)[5] along with the results of this study, suggests that doubling Ng may significantly overestimate actual needs. This is supported by our findings indicating that most regions within the EUCLID domain have a ratio of less than 1.6 ground strike points per CG flash (see Fig. 2b). Therefore, although using a factor of two to estimate Nsg offers a method to enhance lightning protection when only Ng data are accessible, it may also lead to unnecessary expense.

[1]Bouquegneau, C., A. Kern, and A. Rousseau, 2012: Flash density applied to lightning protection standards. Proc. GROUND 2012, Bonito, Brazil, Brazilian Society for Electrical Protection
[2]Rousseau, A. S., F. Cruz, S. Pedeboy, and S. Schmitt, 2019: Lightning risk: How to improve the calculation? Int. Colloquium on Lightning and POower Systems, Delft, Netherlands, CIGRE
[3]International Electrotechnical Commission, 2019: IEC 62858:2019: Lightning density based on lightning location systems – General principles
[4]International Council on Large Electric Systems, 2013: Lightning parameters for engineering applications. Working Group C4.407, CIGRE TB 549
[5]Vagasky, C., R. L. Holle, M. J. Murphy, J. A. Cramer, R. K. Said, M. Guthrie, and J. Hietanen, 2024: How Much Lightning Actually Strikes the United States? Bull. Amer. Meteor. Soc., 105, E749–E759, https://doi.org/10.1175/BAMS-D-22-0241.1.

Line 78. The method includes the CG stroke grouping (flash algorithm) as a first step before calculating the GSP. The flash algorithm parameters used are the classical ones introduced decades ago in the NLDN (Cummins et al., 1998). Since the stroke clustering has a bearing on the further GSP algorithm, I wonder if the EUCLID community has validated the Cummins criteria, since other studies (e.g. San Segundo et al., 2020) suggested that can be adjusted. Since the paper mentions the use of video and E-field records to estimate the EUCLID network detection efficiency, I think they should be used also to calibrate the flash algorithm, previous to the GSP calculation.

The feedback is appreciated. We were not previously familiar with the findings of San Segundo et al. (2020), which indicated that 5-10% of lightning flashes, as observed via LMA, could consist of multiple Ground Strike Points (GSPs) distanced over 10 km apart, and that flashes extending beyond 0.8 seconds are uncommon. By implementing a maximum duration of 0.7 seconds and a radius of 12 km for grouping, there would be a minor increase of 0.5% in the identified number of flashes. While we acknowledge the importance of utilizing the most advanced techniques and parameters in scientific research, it's important to note that our study encompasses various other uncertainties (such as in-cloud to cloud-to-ground misclassification (see also this reviewer's last remark), detection efficiency, etc.) that exert a greater impact than the 0.5% adjustment suggested in San Segundo et al. (2020). Furthermore, the current IEC 62858 standard employs the same algorithm as our study. Therefore, it was decided to maintain the existing flash algorithm in our research.

Regarding the distance between GSPs, note that Poelman et al. (2021)[6] investigated this as well in more detail and found that in Europe (based on ground-truth datasets taken in Austria, France and Spain) the 99[th] percentile of distance between GSP and the first stroke in the flash is below 10 km. However, the suggestion by the reviewer to use video and E-field records to (re)calibrate the flash algorithm is not straightforward. As stated in Poelman et al. (2021)[6] *"It is essential to highlight that the large maximum separation distances could well be the result of a location error by the LLS or a consequence of the manual grouping methodology based on the video information. From the perspective of cloud charge centers and the horizontal extent of downward leaders, it would make more sense to trace the lightning leader back to the location of the preliminary breakdown and only group strokes that emanate from a common charge region. However, this would require observations made by an LMA."* Taken this comment into account, we believe the use of video and E-field measurements is not recommended.

[6]Poelman, D. R., Schulz, W., Pedeboy, S., Campos, L. Z. S., Matsui, M., Hill, D., Saba, M., Hunt, H.: Global ground strike point characteristics in negative downward lightning flashes – part 2: Algorithm validation, *Nat. Hazards Earth Syst. Sci.*, 21, 1921-1933, **2021**, https://doi.org/10.5194/nhess-21-1921-2021

Line 57 "the network's configuration has undergone changes in both the past and during the investigation period, i.e., 2013-2022. However, these changes are not substantial enough to significantly affect the results presented in the following sections" I think this statement needs a supporting reference.

The research by Schulz et al. (2016)[7], analyzed the LA from 2007 to 2014 based on measurements at the Gaisberg Tower. Although changes in the network occurred during this time frame, the LA improved steadily. In the same study, the Detection Efficiency (DE) was evaluated from data spanning different time periods (2008-2012, 2011, and 2012-2013) and regions within the EUCLID network. More recently, EUCLID network performance insights are complemented by the study of Schwalt et al. (2020)[8], which found stroke DE rates to be between 76% and 85.6%, based on ground-truth data

from Austria across the years 2015, 2017, and 2018. The consistency between the DE findings of Schulz et al. (2016) and Schwalt et al. (2020) indicates that stroke DE have remained stable over the years, despite ongoing changes within the network. This stability suggests that network modifications during this period have not significantly affected the outcomes, underlining the reliability of the lightning detection capabilities over time.

[7]Schulz, W., Diendorfer, G., Pedeboy, S., and Poelman, D. R.: The European lightning location system EUCLID – Part 1: Performance analysis and validation, Nat. Hazards Earth Syst. Sci., 16, 595–605, https://doi.org/10.5194/nhess-16-595-2016, 2016.
[8]Schwalt, L., Pack, S., and Schulz, W.: Ground truth data of atmospheric discharges in correlation with LLS detections, Electric Power Systems Research, volume 180, 2020, https://doi.org/10.1016/j.epsr.2019.106065.

Line 122 to 126 It is mentioned that one of the aspects that leads to an overestimation of single-stroke flashes is the "misclassification of IC pulses as CG strokes." Can you develop this statement? What is the relative contribution of this factor? A supporting reference is needed here.

In their 2016 study, Zhu et al.[9] focused on evaluating the classification accuracy (CA) of cloud-to-ground (CG) and intra-cloud (IC) lightning activities, utilizing data from the U.S. National Lightning Detection Network (NLDN) and comparing it with optical and electrical field observations from the Lightning Observatory in Gainesville (LOG), Florida. It was found that the NLDN achieved a CG stroke CA of 92%. For the total of 153 IC events (including isolated IC events, IC events before first return stroke, and IC events after first return stroke), the CA was found to be 86%, while the CA for isolated IC events alone, i.e., complete IC flashes, was notably higher at 95%.
The evaluation of CA for the NLDN is relevant to EUCLID, as both networks employ comparable technology in terms of hardware and software.

[9]Zhu, Y., V. A. Rakov, M. D. Tran, and A. Nag (2016), A study of National Lightning Detection Network responses to natural lightning based on ground truth data acquired at LOG with emphasis on cloud discharge activity, J. Geophys. Res. Atmos., 121, 14,651-14,660, doi:10.1002/2016JD025574